# IgE-Mediated Shellfish Allergy in Children

**DOI:** 10.3390/nu15122714

**Published:** 2023-06-11

**Authors:** Mattia Giovannini, Burcin Beken, Betul Buyuktiryaki, Simona Barni, Giulia Liccioli, Lucrezia Sarti, Lorenzo Lodi, Matteo Pontone, Irene Bartha, Francesca Mori, Cansin Sackesen, George du Toit, Andreas L. Lopata, Antonella Muraro

**Affiliations:** 1Allergy Unit, Meyer Children’s Hospital IRCCS, 50139 Florence, Italy; 2Department of Health Sciences, University of Florence, 50139 Florence, Italy; 3Department of Pediatric Allergy & Immunology, School of Medicine, Acibadem University, 34303 Istanbul, Turkey; 4Division of Pediatric Allergy, Department of Pediatrics, School of Medicine, Koc University, 34450 Istanbul, Turkey; betulbuyuktiryaki@yahoo.com (B.B.);; 5Immunology Unit, Meyer Children’s Hospital IRCCS, 50139 Florence, Italy; 6Pediatric Allergy Group, Department of Women and Children’s Health, School of Life Course Sciences, King’s College London, London SE1 9RT, UK; 7Children’s Allergy Service, Evelina London Children’s Hospital, Guy’s and St Thomas’ NHS Foundation Trust, London SE1 7EH, UK; 8Peter Gorer Department of Immunobiology, School of Immunology & Microbial Sciences, King’s College London, London SE5 9NU, UK; 9Molecular Allergy Research Laboratory, College of Public Health, Medical and Veterinary Sciences, Australian Institute of Tropical Health and Medicine, James Cook University, Townsville, QLD 4811, Australia; 10Tropical Futures Institute, James Cook University, Singapore 387380, Singapore; 11Food Allergy Referral Centre, Department of Mother and Child Health, University of Padua, 35128 Padua, Italy

**Keywords:** allergen, basophil activation test, children, component-resolved diagnosis, cross-reactivity, diagnosis, food allergy, oral food challenge, shellfish, specific IgE, immunotherapy

## Abstract

Shellfish, including various species of mollusks (e.g., mussels, clams, and oysters) and crustaceans (e.g., shrimp, prawn, lobster, and crab), have been a keystone of healthy dietary recommendations due to their valuable protein content. In parallel with their consumption, allergic reactions related to shellfish may be increasing. Adverse reactions to shellfish are classified into different groups: (1) Immunological reactions, including IgE and non-IgE allergic reactions; (2) non-immunological reactions, including toxic reactions and food intolerance. The IgE-mediated reactions occur within about two hours after ingestion of the shellfish and range from urticaria, angioedema, nausea, and vomiting to respiratory signs and symptoms such as bronchospasm, laryngeal oedema, and anaphylaxis. The most common allergenic proteins involved in IgE-mediated allergic reactions to shellfish include tropomyosin, arginine kinase, myosin light chain, sarcoplasmic calcium-binding protein, troponin c, and triosephosphate isomerase. Over the past decades, the knowledge gained on the identification of the molecular features of different shellfish allergens improved the diagnosis and the potential design of allergen immunotherapy for shellfish allergy. Unfortunately, immunotherapeutic studies and some diagnostic tools are still restricted in a research context and need to be validated before being implemented into clinical practice. However, they seem promising for improving management strategies for shellfish allergy. In this review, epidemiology, pathogenesis, clinical features, diagnosis, and management of shellfish allergies in children are presented. The cross-reactivity among different forms of shellfish and immunotherapeutic approaches, including unmodified allergens, hypoallergens, peptide-based, and DNA-based vaccines, are also addressed.

## 1. Introduction

The consumption rates of fish and shellfish have increased because they are known to be a valuable source of protein and omega-3 fatty acids, as well as antioxidants [1,2,3]. The Mediterranean diet, with its wide diversity and richness in fiber and omega-3 fatty acids, has benefits on cardiovascular health (e.g., reducing the rates of coronary heart disease and stroke) [2,4,5]. However, along with a possible higher consumption, the rates of allergic reactions to shellfish may have also increased. 

In this review, epidemiology, pathogenesis, clinical features, diagnosis, and management of shellfish allergies in children are presented. The cross-reactivity among different forms of shellfish and immunotherapeutic approaches, including unmodified allergens, hypoallergens, peptide-based, and DNA-based vaccines, are also addressed.

## 2. Epidemiology

An adverse reaction to seafood has been experienced by approximately 2.5% of the world’s population [1]. Shellfish is one of the most allergenic food groups, which also include milk, egg, peanut, tree nuts, fish, wheat, and soy. In Europe, in a systemic review and meta-analysis, the lifetime prevalence of self-reported shellfish allergy in children aged 2–17 years was found to be 1.3%, and point-prevalence of food challenge-proven shellfish allergy in children aged 6–17 years was reported as 0.08% [6]. The results of a questionnaire-based survey study revealed that the point-prevalence of shellfish allergy in French children aged 2–5 years, 6–10 years, and 11–14 years were 0.2%, 1.8%, and 1.2%, respectively [7].

Shellfish allergy is one of the leading causes of food allergy in most Asian countries, such as Taiwan, Thailand, Singapore, Vietnam, and Hong Kong, where shellfish is more frequently consumed [8].

The EuroPrevall-INCO survey study, including children from Russia, India, and China, determined food allergies in schoolchildren aged 6 to 11 years. Although the percentage of children with shrimp sIgE > 0.70 kUA/L was 13.1% in Shaoguan and 10.3% in India, it was only observed in 4.7% of the subjects in Hong Kong [9]. Survey studies from East Asia presented a parent-reported prevalence of over 5% in children aged 2–5 years in Vietnam [10], 3.4% in all children in Japan [8,11], and 0.84% in schoolchildren in South Korea [12].

Two decades ago, a telephone survey study including adults and children from the United States reported lower shellfish allergy prevalence in children compared to adults (0.5% versus 2.5%) [13]. However, a recent pediatric cross-sectional survey study from the United States revealed a shellfish allergy prevalence of 1.3% in children, showing a potentially increasing trend in shellfish allergy among the pediatric population [14]. In this study, the authors also reported that crustacean allergy was more common than mollusk allergy in the pediatric population [14]. 

The EuroPrevall-iFAAM cohort study, including children aged from 6 years to 10 years from eight European countries (United Kingdom, Germany, Spain, Greece, Netherlands, Poland, Iceland, and Lithuania), yielded a prevalence of parent-reported crustacean allergy as low as 0.2% in primary schoolchildren [15]. These variations in prevalence suggest that several factors, e.g., environmental exposures, dietary habits, and cross-sensitization with other arthropods, such as house dust mites or cockroaches, have effects on the development of shellfish allergy [16]. Nevertheless, methodologies used in these studies might also influence the difference observed in prevalence.

## 3. Classification of Shellfish Species

The term shellfish refers to invertebrates belonging to two different *phyla*: *Arthropoda* and *Mollusca*. Crustaceans are edible arthropods that belong to the *subphylum Crustacea* and the order *Decapoda* [17]. This group includes shrimp, prawn, crab, lobster, krill, crayfish, woodlouse, copepod, and barnacle. Crustaceans are closely related to the arachnid family, which includes, e.g., house dust mites, and insects such as cockroaches [17]. Another large *phylum* of the *Animalia* kingdom is the *Mollusca*, with edible species in three taxonomic classes: *Cephalopoda*, *Bivalvia*, and *Gastropoda* [17] (Figure 1). Crustacean allergy is reported more commonly than mollusk allergy, and shrimps or prawns are the most commonly responsible species for allergic reactions [18,19].

## 4. Adverse Reactions to Shellfish

Adverse reactions to shellfish can be classified into several groups; (1) Immunological reactions, including IgE and non-IgE allergic reactions; (2) non-immunological reactions, including toxic reactions and food intolerance [20]. Toxin-related reactions and food intolerance often resemble clinical manifestations of seafood allergy, such as flushing, or vomiting. However, an appropriate patient workup and diagnostic tests showing sensitization are critical in distinguishing immunological and non-immunological reactions.

### 4.1. Immunological Adverse Reactions

Clinical features of shellfish allergy are IgE and non-IgE-mediated. Typical IgE-mediated reactions occur within about two hours after ingestion and range from urticaria, angioedema, nausea, and vomiting to respiratory signs and symptoms such as bronchospasm, laryngeal edema, and anaphylaxis. A pediatric study including children with shrimp allergy reported that cutaneous clinical manifestations were the most common (70%), and the rate of anaphylaxis was 12% [21]. Adult patients with shellfish allergy are often affected by oropharyngeal signs and symptoms such as swelling of lips, throat tightness, and itchy throat and mouth [22]. On the other hand, both pediatric and adult patients can report self-limited clinical manifestations localized in the oropharyngeal mucosa due to shellfish cross-reactivity with inhalant allergens such as house dust mite (HDM) and tropomyosin (TPM); called mite-shellfish oral allergy syndrome [23]. Inhalant exposure to TPM is considered the primary sensitizer for shellfish allergy [22]. Oral allergy syndrome with shrimp can potentially also be seen in patients undergoing house dust mite oral immunotherapy [24]. Shrimp is also causative for food-dependent exercise-induced anaphylaxis (FDEIA) [25,26]. Akimoto et al. used proteomic analyses to describe P75 homologue and fructose 1,6-bisphosphate aldolase (FBPA) as new potential allergens for shrimp-FDEIA [27]. P75 homologue, as well as myosin heavy chain (MHC), is known as a myofibrillar protein of fast fibers of the crustacean muscles [27].

Rosa et al. [28] reported three adult patients that had allergic reactions after eating shrimp cephalothorax but could tolerate shrimp abdomen. This may be explained by the existence of different allergenic properties of different shrimp species or different body parts of the shrimp.

Non-IgE-mediated reactions, which are being increasingly recognized in children, typically occur several hours or days after allergen exposure and include food protein-induced enterocolitis syndrome (FPIES), food protein-induced enteropathy (FPE) and food protein-induced allergic proctocolitis (FPIAP). In an Italian study conducted with a relatively large number of children with fish/shellfish-induced FPIES, 57 of 70 (81%) patients had reactions exclusively to fish, 9 of 70 (13%) exclusively to shellfish, and 4 of 70 (6%) to both, fish and shellfish [29]. The main features of acute fish and shellfish FPIES compared to other foods, such as cow’s milk or soybean FPIES, were reported as later onset, longer persistence, and the possibility of tolerating fish species other than the offending fish [14,29,30]. Another study showed that children recognize a greater number of epitopes than adults with shrimp allergy [30].

### 4.2. Non-Immunological Adverse Reactions

Contaminating toxins or parasites can also cause adverse clinical reactions to shellfish. Viral and bacterial contamination of shellfish can arise from polluted waters. *Listeria* and *Salmonella* species, as well as viruses such as the *Norwalk* virus, have been implicated. The clinical presentation often includes gastrointestinal signs and symptoms such as vomiting and diarrhea. However, these clinical manifestations usually occur several hours after consumption [31].

Filter-feeding shellfish (mollusk), such as oysters and mussels, can ingest toxic algae. The accumulation of these toxins results in poisoning syndromes in individuals who consume contaminated shellfish (mollusk). Several toxins have different lethal doses, onset and duration times, and various signs and symptoms. Common toxic syndromes are:-Diarrhetic shellfish poisoning; caused by okadaic acid and dinophysis toxins. The clinical manifestations include nausea, vomiting, abdominal pain, and diarrhea [32].-Paralytic shellfish poisoning; caused by saxitoxins which inhibit the generation of action potentials in the membranes of neurons and muscles. Clinical manifestations classically begin with a tingling sensation or numbness of the mouth, neck, fingers, and toes and progress to weakness, limb incoordination, and respiratory difficulty [33].-Neurotoxic shellfish poisoning; caused by brevetoxins that target voltage-gated sodium channels and trigger depolarization of neurons, muscular, and cardiac cells [3]. The signs and symptoms include both neurological (e.g., paralysis and coma) and gastrointestinal clinical manifestations (e.g., nausea, vomiting, and diarrhea) [34].-Ciguatera fish poisoning; caused by the consumption of fish that have accumulated ciguatoxins in their tissues. These toxins target voltage-gated sodium channels, and they can cause gastrointestinal signs and symptoms before or coinciding with neurological and cardiovascular clinical manifestations [35].-Amnesic shellfish poisoning; caused by domoic acid (produced by planktonic diatoms), which targets glutamate receptors in the central nervous system [36]. Usually, gastrointestinal signs and symptoms start first (e.g., nausea, vomiting, diarrhea, and abdominal cramps), and then patients develop neurological clinical manifestations such as confusion, short-term memory loss and coma [37].

## 5. Shellfish Allergens

### 5.1. Tropomyosin

In 1981, tropomyosin (TPM) was identified as a 38-kDa thermostable protein responsible for shrimp allergy [38]. Tropomyosin is a well-known invertebrate pan-allergen that is involved in muscle contraction by interacting with actin and myosin. Other than shellfish, it has been found in numerous invertebrate species such as arachnids (e.g., dust mites), insects (e.g., cockroach), and *Anisakis simplex* [39,40,41]. Tropomyosin has also been described in vertebrates but with non-allergenic properties [42]. The alpha-helical coiled-coil structure provides a highly stable physiological state to TPM. The TPM of invertebrates is known as thermostable and resistant to digestion [17,43,44]. Moreover, different immunoreactivity has been shown in crustacean TPM after heat treatment in several studies [45,46]. Gamez et al. showed reduced IgE-binding capacity of TPM following simulated gastric digestion in a dose- and time-dependent manner [47]. In a study, TPM from shrimp, oyster, and abalone revealed antibody recognition after diet-relevant thermal treatment and peptic digestion, thus, confirming thermostability and resistance against simulated gastric digestion [48].

Reactivity to shrimp TPM Pen a 1 is observed in more than 85% of shrimp-allergic patients [49]. In addition, tropomyosin is also the major allergen in mollusks such as oysters (Cra g 1, Cra g 2), abalones (Hal m 1), snails (Tur c 1) and squid (Tod p 1) [50,51]. A recent multicenter study found that less than 50% of sensitized patients had sIgE to TPM in Italian patients [52]. Shrimp prawn, lobster, and clam TPMs share an amino acid sequence identity of 91–100%. However, the amino acid sequence identity between a crustacean and mollusk TPM is lower, approximately 65% [53,54].

### 5.2. Arginine Kinase

In 2008, arginine kinase (AK) was the second shellfish allergen identified. It was first identified in *Penaeus monodon* (Pen m 2) [55], known as black tiger shrimp, and subsequently in many other crustaceans such as crab [56], octopus [57], cockroach [58], and dust mite [59]. Arginine kinase is less resistant than TPM, and due to its thermolability and volatility, it can be considered responsible for respiratory signs and symptoms and as an occupational allergen [60]. Since inhaled bioaerosols containing seafood allergens can induce allergic reactions, workers engaged in the seafood industry, food preparation (e.g., chefs and waiters in restaurants), and harvesting (e.g., fishermen, aquaculture) are mostly at risk of occupational allergy [61]. To date, the percentage of patients sensitized to prawns who recognize AK is not well defined; however, it is believed to range between 10% and 51% [30,62].

### 5.3. Myosin Light Chain

In 2008, myosin light chain (MLC) was identified in an American white shrimp, *Litopenaeus vannamei* (Lit v 3) [63], and later in lobster [64], crab [65], and cockroach [66]. It is considered a minor allergen resistant to heat processing [44]. In an Italian study including shrimp-allergic patients, shrimp-tolerant patients, and healthy controls, TPM was found to be the most frequently recognized allergen alone (12.1%) and in combination with sarcoplasmic calcium-binding protein (SCP) (31%) or MLC (36.2%) in allergic patients and these three allergens were suggested to be related to a positive food challenge outcome [67].

### 5.4. Sarcoplasmic Calcium-Binding Protein

In 2008, sarcoplasmic calcium-binding protein was first described in *Penaeus monodon* (Pen m 4) [68]. It is characterized as a highly resistant and stable protein and has a high sequence identity among crustaceans but a low identity between crustaceans and mollusks [69,70]. Although considered a minor allergen, it can be clinically relevant regardless of sensitization to TPM [62]. Interestingly, SCP sensitization was found to be more common in children (73%) compared to adults (10%), suggesting that it is an important allergen in the pediatric population [69].

### 5.5. Troponin C

Troponin C (TpC) is a 20 kDa protein with unknown heat stability. It has been characterized in shrimp and also cockroach [71]. Green crab and lobster troponins share approximately 50% to 60% identity with shrimp TpC [67].

### 5.6. Triosephosphate Isomerase

Triosephosphate isomerase (TIM) was characterized in shrimps, crayfish, and cockroach in 2009 [66]. Its molecular weight is approximately 28 kDa and is probably heat labile. Five of eight (63%) shrimp-allergic patients had IgE binding to Cra c 8 in immunoblotting and 7/31 (23%) shrimp-allergic sera had positive results to Cra c 8 [66]. Nevertheless, further studies are needed to understand the cross-reactivity of TIM among several invertebrate species.

### 5.7. Other Allergens

Other reported shellfish allergens are paramyosin, fatty acid-binding protein, hemocyanin, myosin heavy chain, α-actine, smooth endoplasmic reticulum Ca^+2^ ATPase, glyceraldehyde-3-phosphate dehydrogenase, ovary development-related protein, troponin I [72].

The clinical relevance of these allergens is still unclear. However, hemocyanin seems to have an important role in cross-reactivity with mites, cockroaches, and other invertebrates such as snails [64] [73,74].

The list of known shellfish allergens is given in Table 1. Allergen sensitization rates are also presented [17,75].

### 5.8. Cross-Reactivity

The invertebrate TPM is a pan-allergen that is heat stable and known to be implicated in cross-reactivity among crustaceans, mollusks, HDM, and cockroaches due to its high amino acid sequence identity among invertebrates [17,88].

#### 5.8.1. Cross-Reactivity among Shellfish Species

Tropomyosin among crustacean group demonstrates a strong cross-reactivity due to high amino acid identity, over 95% among prawns, crabs, and lobsters [67].

There is very limited knowledge about TPMs in the mollusk group. The major allergens from various mollusk species, such as mussels, abalone, oyster, squid, and cockle, share amino acid identity between 65 and 99% [89]. A study by Kamath et al. [90] demonstrated in a murine model that mollusk TPM could independently elicit a strong IgE response, primarily due to TPM, without any prior sensitization to crustacean allergens. Vidal et al. [91] reported that 17 of 31 (54.8%) subjects with crustacean anaphylaxis were tolerant to mollusks. Cox et al. [92] reported that about 75% of patients with a crustacean allergy react to more than one type, but less than 50% react to mollusks. Conversely, >70% of those with mollusk allergy are at risk of reacting to crustaceans. Among mollusk-allergic patients, approximately 50% report reactions to more than one species of mollusk. Furthermore, approximately 10–15% of patients allergic to any shellfish are allergic to both crustaceans and mollusks. Our knowledge of true clinical cross-reactivity is insufficient due to the lack of clinical studies. Patients with an allergy to any shellfish may avoid consuming a different shellfish, so studies performing oral food challenges (OFCs) to various shellfish species would be more informative in terms of reporting true clinical cross-reactivity among this group.

#### 5.8.2. Cross-Reactivity between Shellfish and Fish

The major allergen in fish allergy, parvalbumin, is distinct from those in shellfish [93]. Therefore, cross-reactivity between them is not noteworthy. In a retrospective study, at least 21% of those with fish allergy were allergic to crustaceans [94].

Nevertheless, TPM has also been identified in fish-allergic patients, 32% of fish-allergic children were sensitized to TPM from salmon and Asian seabass [95]. Interestingly, shrimp TPM showed low cross-reactivity to TPM fish despite a high sequence similarity in two studies by Xu et al. [96,97], which needs further investigation.

#### 5.8.3. Cross-Reactivity between Shellfish, HDM, and Cockroach

It is well-known that TPMs from HDMs and TPMs from shellfish share a high amino acid sequence homology, with an 81% amino acid sequence similarity between HDMs and prawns [98], and a high sequence homology to TPMs, with an 82% similarity between prawns and cockroach [99]. 

Tropomyosins of mites and cockroaches have a high sequence identity to shrimp Pen a 1, 78.5–81.7% and 82.4%, respectively. Eight IgE-binding epitopes were identified in Pen a 1 from *Penaeus aztecus,* and these new epitopes were proposed to be the major IgE epitopes [100]. The multiple sequence alignment of shrimp (Pen a 1, Pen m 1), crab (Por p 1), lobster (Hom a 1), and HDM (Der p 10 and Blo t 10) TPMs revealed that Pen m 1, Pro p 1, and Hom a 1 have almost identical sequences at all the eight identified Pen a 1 IgE epitopes. The sequence identity of the HDM TPMs Der p 10 and Blo t 10 to these eight IgE epitopes of Pen a 1 was also high (>80%) [54,101].

#### 5.8.4. Cross-Reactivity between Shellfish and *Anisakis simplex*

*Anisakis simplex* is a parasitic nematode, which mostly infects fish but can also infect shellfish. Cross-reactivity between Anisakis, insects, mites, and crustaceans is thought to be caused by tropomyosins. Unfortunately, the prevalence data regarding cross-reactivity is not known due to the lack of large cohort studies [102].

#### 5.8.5. Tropomyosin IgE Cross-Reactivity between Shellfish and Edible Insects

Edible insects are currently an increasing dietary component due to their high protein content and low biomass, water, and energy inputs during the production process. Silkworms, grasshoppers, locusts, mealworms, crickets, butterflies, moths, cicadas, and dragonflies are some of the edible insects. With the increasing consumption of these insects, allergic reactions have been reported, especially in Asian countries where these insects are a great part of the cuisine. Recently, Broekman et al. [103] reported four cases who developed allergic respiratory signs and symptoms during professional or domestic mealworm breeding. The domestic breeders also reported food allergic clinical manifestations. One individual was sensitized to house dust mite, and one had shrimp sensitization, but all of them could consume shrimp without any allergic reaction. Mealworm was thought to be the primary sensitizer also in the patient with HDM sensitization.

Studies have demonstrated shellfish-allergic patients to be cross-reactive to insects and house dust mite via TPM [104]. Jenkins et al. studied the sequence of human and shrimp TMs and established that proteins with a sequence identity to a human homolog above approximately 62% were rarely allergenic [105]. Cross-reactivity has been reported rarely below 50% sequence identity and more commonly above 70% [106]. Palmer et al. [107] investigated the cross-reactivity of different insects (e.g., mealworm, superworm, waxworm) to house dust mites and shrimp and reported that mealworm, waxworm, and superworm showed lower IgE binding compared to other insect species.

## 6. Diagnosis

The diagnostic workup for IgE-mediated shellfish allergy includes a thorough review of clinical history alongside a skin prick test (SPT) and/or serum-specific IgE (sIgE) measurement.

Despite the ongoing advancements of different diagnostic techniques, achieving a diagnosis with a single-step test is not easy. Hence, a stepwise approach to conclude the shellfish allergy diagnosis is advised (Figure 2) [108,109].

The current workflow is to perform allergy testing, including SPT and blood sIgE tests, for patients presenting with a clinical history of shellfish allergy. Further tests include component-resolved diagnosis (CRD), basophil activation tests (BAT), and IgE-crosslinking-induced luciferase expression (EXiLE) tests. Although sIgE shows the sensitization to whole extract, component-resolved diagnosis (CRD) measures sIgE against individual components involved in IgE-mediated reactions. Basophil activation tests (BAT) and IgE-crosslinking-induced luciferase expression (EXiLE) tests show the biological activity compared to SPT and sIgE, which measure the IgE sensitization. Therefore, if possible, CRD, BAT, and EXiLE tests should be performed following SPT and sIgE tests. OFCs are still the most accurate way of detecting clinical allergies. However, these tests are time-consuming, labor-intensive, expensive, and have the risk of severe life-threatening reactions.

### 6.1. Clinical History

Clinical history is the mainstay of the diagnosis of shellfish allergy. Although most of the patients experience signs and symptoms upon ingestion, e.g., skin contact, or inhalation of aerosolized allergens during cooking or boiling may cause allergic reactions too.

Signs and symptoms occur within about 2 hours upon ingestion of the allergenic food. Other data, e.g., concerning the age of onset, type of shellfish, type of clinical manifestations, severity, the time interval between the exposure and the occurrence of reactions, prior history of a similar reaction, food preparation techniques (e.g., raw, cooked, boiled), amount of the allergen, route of exposure, cofactors (e.g., exercise, illness, medicines), other allergic conditions including cockroach and house dust mite allergy as well as family history of atopy should be obtained during the evaluation of the patient [110].

Cross-reactivity has been reported in >75% of shellfish-allergic patients [108]. Due to allergen sequence homology, patients allergic to one type of crustacean may also react to other crustaceans. Of note, patients with crustacean allergy do not always react to mollusks, which may be an alternative protein source. In a previous study including children with shrimp allergy, cross-sensitization for other crustaceans and mollusks was found to be 57% and 26%, respectively [21]. In general, an allergic workup is recommended for different types of shellfish allergens if the patient does not consume these allergens safely. On the other hand, if a type of shellfish is tolerated, testing for this type is not required.

In the differential diagnosis of shellfish allergy, it is also important to consider non-adverse immunologic reactions, which occur later than about 2 hours due e.g., to toxins and parasites and present with signs and symptoms including gastrointestinal and/or neurologic clinical manifestations.

Following a detailed history, SPT and/or sIgE are performed as first-line diagnostic tests.

### 6.2. Skin Prick Test

Skin prick tests (SPT) have been commonly used since the first description in 1924 due to their ease of use, low cost, minimal invasiveness, and rapid results [111]. A drop of allergen is introduced to the epidermis on the patient’s forearm’s volar surface or upper back using a lancet or commercial test device. Histamine and normal saline are used as positive and negative controls, respectively. A wheal diameter equal to or greater than 3 mm with a negative control following 15–20 minutes of application is considered a positive test result. Although not common, an allergic reaction may occur during the procedure. Tests can be performed by applying commercial whole allergen extracts, or, by fresh allergens (prick-to-prick tests (PTP)). However, some factors must be taken into consideration when analyzing the results, such as the likelihood of cross-reactivity among shellfish, house dust mites, and cockroaches, the variability of test protocols used, the lack of standardization in allergen preparations used in the test, and the fact that the shelf-life and the stability of the reagent can interfere with the sensitivity and specificity of these conventional tests [109]. In a study including children and adults, five commercial shellfish SPT extracts reported a high variability of IgE reactivity in immunoblotting [112], with a sensitivity range of 59–79%. The authors also observed an interesting loss of protein bands in commercial extracts compared to freshly prepared in-house shrimp extract during sodium dodecyl-sulfate polyacrylamide gel electrophoresis (SDS-PAGE). Several studies attempted to determine cut-off values of SPT to shellfish for predicting clinical reactivity. In Thailand, children with shrimp allergy were subjected to SPT using extracts of seawater shrimp *Penaeus monodon*, Pm) (PmSPT), freshwater shrimp (*Macrobrachium rosenbergii*, Mr) (MrSPT), commercial shrimp (ComSPT), and prick-to-prick (PTP) tests (PmPTP, MrPTP) and underwent oral food challenges [113]. SPT using crude extracts and PTP were found more useful than commercial extracts for screening the sensitization to shrimp. In children with Pm allergy, PmSPT of 30 mm yielded an 80% positive predictive value (PPV) for clinical reactivity. In patients with Mr allergy, MrSPT of 30 mm provided 95% PPV [113].

Among the factors that could affect SPT, storage conditions and thermal processing have been investigated in some studies to provide optimal conditions for reliable results. Piboonpocanun et al. [114] suggested that shrimp extracts can be stored at −20 °C for 4 weeks since they observed loss of allergenicity in extracts at 4 °C. Later, Pariyaprasert et al. [115] investigated the stability and potency of raw and boiled shrimp extracts used in SPT at different time points. Raw and boiled shrimp extracts were found stable at 4 °C for 30 days and induced ≥10 mm mean wheal diameter response in allergic patients, which was also comparable with prick-to-prick to fresh shrimps. The authors observed that the wheal sizes of boiled *Penaeus monodon* extracts were smaller than the raw extracts, but there was no difference in SPT wheal sizes between raw and boiled extracts of *Macrobrachium rosenbergii*. In contrast, Carnes et al. suggested that using boiled extracts was more effective in the diagnosis of shellfish allergy [45]. The authors reported that boiled extracts induced larger wheal sizes and a higher percentage of SPT reactions than raw extracts. The discrepancies in the results may be caused by methodological differences and address the need for standardized extracts for reliable results.

### 6.3. Specific IgE

Specific IgE can be measured in vitro with different diagnostic methods such as ImmunoCAP (Phadia/Thermo Fisher Scientific, Uppsala, Sweden), IMMULITE (Siemens Healthcare Diagnostics, Los Angeles, CA, USA), HYTEC-288 (Hycor-Agilent, Garden Grove, CA, USA), and ALEX (Macroarray Diagnostics, Wien, Austria). However, commercially available allergens are limited, a drawback in clinical practice. Although shrimp sIgE is a useful test for detecting sensitization, higher levels do not correlate with the severity of the reaction, and patients may experience anaphylaxis even with very low levels of sIgE. Moreover, children with a cockroach or house dust mite sensitization may have higher levels of sIgE, which may not reflect clinical reactivity [108]. The sensitivity and specificity of whole shellfish extracts are not found to be high, and the predictive value varies depending on the prevalence of the population. In an Asian study on adult and pediatric patients, the sensitivity and specificity of shrimp sIgE were 62% and 50%, respectively, for diagnosing shrimp allergy [22].

### 6.4. OFC

There are three types of OFCs: open, single-blind, and double-blind. A double-blind placebo-controlled oral food challenge (DBPCFC) remains the gold standard diagnostic method for shellfish allergy, as in all food allergies. However, OFCs are time-consuming, labor-intensive and expensive tests with a risk of anaphylaxis, so clinicians should assess whether an OFC (with specific doses and timing) is necessary to confirm the diagnosis [116].

In clinical practice, open OFCs are suitable for patients with consistent history. Although OFCs are usually performed to confirm or rule out the diagnosis of the culprit allergen, they may be performed to introduce a cross-reactive food in the children’s diet since a multicenter Italian study on adult and pediatric patients showed that the cross-reactivity between crustaceans and mollusks is not adequately predicted by the available diagnostic methods [117].

During OFCs, increasing amounts of shellfish are gradually given until an age-appropriate portion is reached. Although 5 mg of shellfish protein was proposed as an initial dose previously, in recent studies, 3 μg is now recommended as the first dose of OFCs to minimize the risk [118,119]. As there is a possibility for severe reaction during OFCs, threshold dose distributions of different food allergens have been determined in the European population who underwent DBPCFC tests to remain safe as far as possible. Although estimated doses eliciting reactions in 10% of the allergic population (ED10) for peanut, hazelnut, and celery were 2.8 mg, 9 mg, and 1.6 mg, respectively, for shrimp, the dose distributions were different with an ED10 of 2.5 g of protein [119].

The shellfish extract can be masked in chocolate pudding or burgers minced with chicken meat and herbs. The EuroPrevall Study, conducted in 12 outpatient clinics across Europe, aimed to improve food allergy diagnosis and management [120]. In the study, shrimp was blinded in a burger including chicken meat, dried oregano, dried onion, and ground black pepper, with a maximum amount of 16 g (equivalent to 3 g protein). The DBPCFC tests were performed in seven doses, including 3 mcg, 60 mcg, 600 mcg, 12 mg, 120 mg, and 1 g and 3 g shellfish protein [120]. If there were no signs or symptoms, an open challenge was performed, and two doses of cooked shrimp of 30 and 50 g, equivalent to 6 and 9 g of protein, were given. The cumulative dose of shrimp in this study was 102 g. In children, the test could be stopped when the age-appropriate doses were given.

### 6.5. Component-Resolved Diagnosis (CRD)

In recent years, the discrepancies in the results of conventional tests and the aim to reduce the need for OFCs increased the interest in new diagnostic methods. Component-resolved diagnosis (CRD), which allows specific detection of sIgE reaction to individual allergenic molecules, provides more information regarding the IgE recognition profile.

Yang et al. observed that the specificity of TPM sIgE (92.8%) was greater than SPT with commercial extract (64.2%) and sIgE to shrimp (75%) for predicting the clinical reactivity to shrimp [121]. In a study by Gamez et al. [122], sIgE to shrimp component, recombinant TPM rPen a 1, was detected by 98% of shrimp-allergic patients. Pascal et al. [67] evaluated the diagnostic values of IgE recognition against shrimp allergens, showing a that the specificity of TPM epitopes can reach 100% specificity but with variable sensitivity of 33–86%. Although TPM and sarcoplasmic-calcium-binding-protein sensitization were related to a positive oral food challenge outcome, arginine kinase and hemocyanin appeared to be cross-reacting allergens [58]. Myosin light chain sensitivity was also found to help show clinical reactivity [58]. All these works were carried out on adult and pediatric patients.

Sensitization to allergen components can be measured using singleplex or multiplex assays. A study evaluated the clinical utility of singleplex (ImmunoCAP) and multiplex (ImmunoCAP ISAC) methods for the diagnosis of shrimp allergy in adult patients. ImmunoCAP detected elevated levels of shrimp sIgE in all patients. However, ISAC 112 indicated only a 50% detection rate against at least one out of three shrimp allergen components [123].

### 6.6. Basophil Activation Test (BAT) and IgE-Crosslinking-Induced Luciferase Expression (EXiLE)

The Basophil activation test (BAT) is an in vitro functional assay used for food allergy diagnosis. Nevertheless, the routine use of BAT is limited by, e.g., its cost, reproducibility, and the short-living nature of basophils. BAT measures the percentage of activated basophils (the expression of activation markers such as CD 63 and/or CD 203 on basophils) in response to an allergen [109].

Emerging IgE-crosslinking-induced luciferase expression (EXiLE) is a relatively new method similar to BAT that measures specific allergen-IgE crosslinking without the need for procedures such as flow cytometry-based analyses. In this assay, a rat basophilic leukemia (RBL) cell line is transfected with human IgE receptor FcεRI a/b/g-subunits and luciferase reporter gene (RS-ATL8). Crosslinking of FcεRI induces the expression of nuclear factor of activated T-cells (NFAT) and, in turn, regulates the expression of the luciferase reporter gene [124]. Therefore, the luciferase signal reflects the degree of IgE crosslinking.

The value of EXiLE in shrimp allergy was shown by Jarupalee et al. [125] who published that the 38 kDa- and 115 kDa shrimp protein fractions induced higher reporter signals by EXiLE test. Wai CY et al. [126] recently reported the diagnostic values of BAT and EXiLE compared to conventional allergy tests. Thirty-five shrimp-allergic and tolerant subjects, defined by DBPCFC, were assessed in this study, comparing the diagnostic accuracy of conventional SPT, shrimp extract, and rPen a 1 sIgE measurement with BAT and EXiLE. Both assays showed a significant association, but BAT gave superior diagnostic power with 87% sensitivity, 94% specificity, 93% PPV, and 89% NPV. This study also highlighted that BAT may be better than SPT and sIgE measurement in shrimp allergy diagnosis.

## 7. Management

To date, clinical management of shellfish allergy is still challenging. Patients are advised to avoid shellfish and use their rescue medication in the case of an allergic reaction. Prescription of adrenaline auto-injectors and training, e.g., on how and when to use them is recommended in selected cases according to international guidelines. Compared to other food allergies such as cow’s milk and egg, the development of tolerance is less common in shellfish allergy in children, which may persist throughout life [89]. For the supplementation of omega-3 fatty acids, individuals regularly take fish oil, which may contain traces of shellfish proteins. Even though the risk of allergic reactions to fish oils is considered to be very low in individuals who are allergic to shellfish, it is recommended that these patients should seek medical advice about the risk of fish oil before consumption [127]. To prevent allergic reactions, it is recommended to prescribe fish free omega-3 [128].

Food allergen immunotherapy has gained importance as a specific treatment that improves the quality of life of the affected individuals and reduces the severe reactions from accidental exposures [129]. However, allergen-specific immunotherapies (AITs) for shellfish allergy have been investigated in models but have not been implemented in general practice yet.

### 7.1. AIT

#### 7.1.1. Shrimp Extract

Studies investigating the utility of allergen extracts for managing shellfish allergy are scarce. Refaat et al. [130] evaluated the efficacy of sublingual immunotherapy with shrimp extract in Egyptian shrimp-allergic adult and pediatric patients (who also had asthma, allergic rhinitis, or chronic urticaria) and healthy controls. They found a significant reduction in allergic signs and symptoms and shrimp sIgE levels. However, sIgE to two shrimps (*P. semisulcatus* and *M. stebbingi*) were elevated after 6 months of starting immunotherapy. The subgroup analysis revealed that immunotherapy was more effective in patients with allergic rhinitis compared to asthmatics or those with chronic urticaria. In addition, the AIT was also safe and well tolerated.

Nguyen DI et al. [131] recently published the data of 3 patients who received omalizumab (a humanized monoclonal anti-IgE antibody)-facilitated oral immunotherapy (OIT) for shrimp allergy in a multiple OIT trial. All 3 patients in this clinical trial were able to reach a maintenance dose of 1g, and 2 out of 3 had no reaction with the 12 g (equivalent to approximately 3 medium-sized white prawns) DBPCFC dose at week 30. Shrimp OIT seems to be an efficacious treatment. However, the sample size of these studies was very small, and there is scarce published literature so far on e.g., the optimal shrimp allergen product that should be used, the escalation regimen, the appropriate maintenance doses, and the role adjunct therapies, such as omalizumab, have to achieve desensitization. Combining omalizumab and oral immunotherapy aims to reduce adverse events and facilitate rapid up-dosing [132].

#### 7.1.2. Shrimp Allergen TPM Met e 1

Leung et al. [133] investigated the efficacy and safety of the major cross-reactive allergen, *Metapenaeus* TPM Met e 1, in a murine model of shrimp hypersensitivity. Low (0.01 mg), medium (0.05 mg), or high (0.1 mg) rMet e 1 doses were applied through the intraperitoneal route. All doses were successful in desensitization with a significant decrease in sIgE. Nevertheless, significant up-regulation of regulatory genes, such as transforming growth factor beta (TFG-β), interleukin-10 (IL10), and forkhead box P3 (Foxp3), were only developed with the low and medium-dose treatments. Notably, one of the six mice in the high-dose group was found dead immediately after the first dose of AIT. Tropomyosin has exceptional IgE crosslinking capacity due to its secondary coil-coiled structure with well-spaced epitopes that might be related to the severe AIT adverse events of TPM-based AIT [134].

#### 7.1.3. Peptide-Based Immunotherapy

Peptide-based immunotherapy has also been studied in shrimp allergy. Ravkov et al. first identified 28 TPM peptides and 17 T cell-specific epitopes by proliferation and cytokine release assays. Subsequently, Wai et al. [135] identified T cell epitopes of *Metapenaeus* TPM Met e 1 from 18 synthetic peptides and treated Met e 1-sensitized mice with six major Met e 1 T cell epitopes twice a week for four weeks. They demonstrated that peptide-based shrimp immunotherapy was capable of alleviating allergic responses by restoring the Th1/Th2 balance, generation of IgG antibodies against Met e 1, and enhancement of regulatory T (Treg) cells responses. They also found a reduction in the recruitment and activation of effector cells such as mast cells, eosinophils, and goblet cells.

#### 7.1.4. Hypoallergens

Reese et al [136] conducted the first study involving hypoallergenic TPM (Pen a 1 mutant VR9-1) by mutating 12 positions deemed critical within the IgE major epitopes [136]. In RBL histamine release assay, the VR9-1 showed a 10- to 40-fold reduced allergenic potency compared to wild-type Pen a 1. Two hypoallergenic versions of shrimp TPM, MEM49 and MED171 were also constructed by substitution of the IgE-binding epitopes of Met e 1 [137]. Both MEM49 and MED171 showed a marked reduction in IgE allergenicity, as well as ability to induce blocking IgG antibodies [137].

#### 7.1.5. DNA Vaccine-Based Immunotherapy

DNA vaccine-based immunotherapy is another emerging therapy for shrimp allergy. Wai CYY, et al. [138] designed MEM49 and MED171 as shrimp hypoallergens and investigated the effectiveness of hypoallergen-encoding DNA vaccines to treat TPM-induced shrimp allergy. It was shown that intradermal treatment of mice with these two vaccines effectively down-regulated systemic allergic clinical manifestations, TPM-specific IgE level, Th2 cytokine expression, and inflammatory responses, and up-regulated the level of IgG2 antibodies and Treg cells.

## 8. Conclusions

Shellfish allergy is one of the most common food allergies worldwide. There is increasing literature on its diagnosis and management. However, data concerning the pediatric population is lacking. Improving diagnostic accuracy is essential, and in vitro assays represent promising tools for this purpose. Nevertheless, studies with large cohorts are needed before incorporating them into the clinics. Over the past two decades, the knowledge gained on the identification of the molecular features of different shellfish allergens improved the diagnosis and the potential design of allergen immunotherapy for shellfish allergy. Unfortunately, immunotherapeutic studies and some diagnostic tools are still restricted in a research context and need to be validated before being implemented into clinical practice. However, they seem promising for improving management strategies for shellfish allergy.

## Figures and Tables

**Figure 1 nutrients-15-02714-f001:**
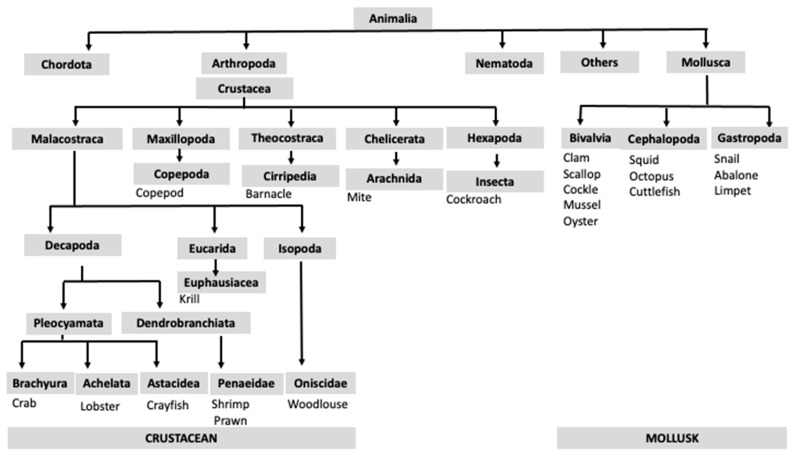
Taxonomic classification of shellfish species within the *Crustacea subphylum* and the *Mollusca phylum*. Modified from [17].

**Figure 2 nutrients-15-02714-f002:**
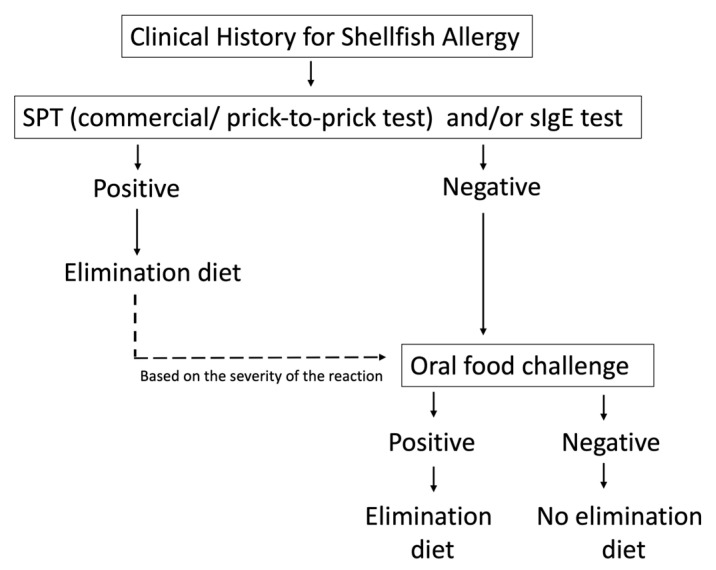
Proposed diagnostic workup for IgE-mediated shellfish allergy in children. (SPT: Skin prick test, sIgE: specific Immunoglobulin E). Modified from [93].

**Table 1 nutrients-15-02714-t001:** List of shellfish allergens according to the International Union of Immunological Societies (IUIS) Allergen Nomenclature. Modified from [71].

Biochemical Name	Molecular Weight	Heat Stability	Route of Exposure	Physiological Function	Sources (Examples)	Allergen	IgE Sensitization (%)	References
Tropomyosin	33–38 kDA	Stable	Ingestion Inhalation	Binds to actin and regulates the interaction of troponin and myosin	Shrimp Lobster Crab Octopus SnailWhelk Abalone Clam Mussels	Pen a 1 Lit v 1 Pen m 1 Hal m 1 Cra c 1 Mel l 1 Pan b 1 Pen i 1 Met e 1 Por p 1 Hom a 1 Scy o 1 Scy p 1 Scy s 1 Cha f 1	72–98	[76,77,78,79,80]
Arginine kinase	38–41 kDA	Labile	Ingestion Inhalation	Catalyzes the reversible transfer of phosphoryl group from ATP to arginine	Shrimp Crab Octopus	Pen a 2 Pen m 2 Cra c 1 Lit v 2 Scy o 2 Scy p 2 Scy s 2 Cha f 2 Met e 2 Por p 2	10–51	[55,81]
Myosin light chain	17–20 kDA	Stable	Ingestion	Regulates smooth muscle contraction	Shrimp Lobster	Pen m 3 Lit v 3 Cra c 3 Hom a 3	19–55	[63,82]
Sarcoplasmic calcium-binding protein	20–25 kDA	Stable	Ingestion	Acts as a calcium buffer regulating calcium-based signalling	Shrimp	Pen m 4 Lit v 4 Cra c 4 Mel l 4 Pon l 4 Scy p 4 Cha f 4 Met e 4	29–50	[68,69]
Troponin C	20–21 kDA	Unknown	Ingestion	Regulates interaction of actin and myosin during muscle contraction	Shrimp Lobster	Lit v 6 Cra c 6 Hom a 6 Pen m 6 Scy o 6 Pan b 6	12–29	[66,83]
Triosephosphate isomerase	25 kDA	Labile	Ingestion Inhalation	Catalyses conversion of dihydroxyacetone phosphate to glyceraldehyde 3-phosphate in glycolysis	Shrimp	Pen m 8 Cra c 8 Arc s 8 Pro c 8 Scy p 8	15–23	[66]
Paramyosin	99 kDA	Unknown	Ingestion	Functions as a cytoplasmic protein that plays an essential role in the processes of myoblast fusion	Octopus Abalone Turban Shell Mussels	Myt g PM Oct v PM	* NR	[83]
Fatty acid-binding protein	15 kDA	Stable	Ingestion	Coordinates lipid trafficking and signalling in cells		Lit v 13	10.3	[84]
Hemocyanin	72–75 kDA	Stable	Ingestion	Binding, transportation, and storage of dioxygen within the blood of many invertebrates	Shrimp	Lit v 1 Hemocyanin Pan b Hemocyanin Mac r Hemocyanin	29–47	[85]
Myosin heavy chain	225 kDA	Unknown	Ingestion	Muscle contraction	Shrimp Snail	Pan b Myosin	* NR	[60]
α-actine	31–42 kDA	Unknown	Ingestion	Muscle contraction	Shrimp		* NR	[60,81]
Smooth endoplasmic reticulum Ca^+2^ ATP ase	113 kDA	Unknown	Ingestion	Enzyme	Crab	Chi o SERCA	* NR	[81]
Glyceraldehyde-3-phosphate dehydrogenase	37 kDA	Unknown	Ingestion	Enzyme for anaerobic glycolysis	Shrimp		* NR	[60]
Ovary development-related protein	28 kDA	Unknown	Ingestion	Ovary development	Crab	Eri s 2	* NR	[86]
Troponin I	30 kDA	Unknown	Ingestion	Calcium-binding protein	Crayfish	Pon I 7	* NR	[87]

* NR: Not reported.

## Data Availability

Not applicable.

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
