# Peer review of "IgE-Mediated Shellfish Allergy in Children"

_nutrients, 2023, doi:10.3390/nu15122714_

Round 1

Reviewer 1 Report

General Comment

The work of Mattia Giovannini, Burcin Beken, Betul Buyuktiryaki, Simona Barni, Giulia Liccioli, Lucrezia Sarti, Lorenzo Lodi, Matteo Pontone, Irene Bartha, Francesca Mori, Cansin Sackesen, George du Toit, Andreas Ludwig Lopata, Antonella Muraro, entitled: “IgE-mediated Shellfish Allergy in Children” has reviewed the epidemiology, pathogenesis, clinical features, diagnosis, and management of shellfish allergies in children. The timing, scope an theme of the review is justifiable and warranted. The style and grammar of the English language is appropriate and allows easy follow up, aside of few the few terms and expressions that need to be adequately amended and elaborated. Please attend moderate and minor comments specified down below, with some of them intending to increase the comprehensiveness and clarity of the proposed review.

Comments

1.       Lines 67-70. Please rewrite more precisely, taking care of sensible comparison when addressing results of reference 6. “In a survey in which skin prick tests (SPTs) and specific IgE (sIgE) were used to determine food allergy prevalence, 13.1 % and 10.3% of the Chinese and Indian populations, respectively showed sIgE that exceeded 0.70 kUA/L; however, only 2% of the Hong Kong population was positive to shrimp by SPT [6]”. It cannot be understood properly, because in the first part of the sentence skin prick tests (SPTs) and specific IgE (sIgE) are mentioned and it seems that prevalence are based on 2 sources simultaneously (Chinese and Indian), while for Hong Kong population only SPT is reported as a factor based on what prevalence is determined. Finally when reading this reference 6 and its table III, it was reported that Hong Kong population prevalence is 4,7% while Guangzhou is 2%. Please correct.

2.       Line 76 Correct typos. There is a “.” Before reference [11]

3.       Lines 79-80. “In this study, the authors also reported that crustacean allergy is more common than mollusk allergy in the pediatric population.” Please cite this study. As authors might be aware, presentation of new, unpublished results in the review article is discouraged (as per MDPI directions).

4.    Lines 84-87. “These variations in prevalence suggest that environmental exposures, dietary habits, and cross-sensitization with other arthropods, such as house dust mites or cockroaches, have effects on the development of shellfish allergy.”  Please elaborate more on possible variations among prevalence reported. For example, it could be due to different methodologies used.

5.       Line 95 Correct typos. There is a “.” Before reference (Figure 1)

6.    Explain on what P75 is reffered to? “p75(NTR) signaling provides a prosurvival response in vestibular schwannoma cells by activating NF-kappaB independent of JNK” or something else?

7.    Elaborate more on Non-IgE-mediated reactions. For example, outline the most frequent ones. Such as Non-IgE-mediated gastrointestinal food allergic diseases (non-IgE-GI-FA), which are being increasingly recognized in children, consist of three main entities: food protein-induced enterocolitis syndrome (FPIES), food protein-induced enteropathy (FPE) and food protein-induced allergic proctocolitis (FPIAP).

8.    Tropomyosin is a pan-allergen that is involved in invertebrate muscle contraction.” The sentence should be rewritten to something like this:” Tropomyosin is a renowned invertebrate’s pan-allergen that is involved in muscle contraction in both invertebrates and vertebrates.”

9.       Line 175. The TPM of invertebrates is thermostable and resistant to digestion [40, 41, 14].” Please elaborate with more relevant references employing INFOGEST protocol that simulates physiological conditions of human digestion. Also the newer ones. For example: Comparative digestion of thermally treated vertebrates and invertebrates allergen pairs in real food matrix.” DOI: 10.1016/j.foodchem.2022.134981

10.   Consider to place the source of the info within the caption of Figure 2. E.h. which health panel/guides has proposed the diagnostic configuration shown.

11.   Lines 459-460. Correct typos and correct the statement that BAT is emerging tool. I disagree that BAT is emerging tool. It has been used in Karolinska Institute for more than 15 years as a tool to address food allergies.

Kind regards

The style and grammar of the English language is appropriate and allows easy follow up. It should be only checked for minor spelling mistakes.

Reviewer 2 Report

Some suggestions and corrections can be found in the document "comments to the review.pdf"

Reviewer 3 Report

Overall, this review provides interesting information on a popular subject of allergy and atopy. Some discussion is provided on types of allergens, as well as diagnosis and treatment. The sections detailling allergens and categorisation are rather superficial, and could be better supplemented with figures and some summary tables, which are rather lacking. Considering it is a review article, more comprehensive information rather than arbitrarily selected statements would provide a more cohesive narrative. With regard to the sections on diagnosis and treatment, there is lack of depth regarding mechanisms and current / future clinical practice, so these sections should be addressed before publication. 

Quality of English is acceptable. Some minor grammatical and spelling errors which should be addressed before publication.
